# Nano-ZnO-Induced Drought Tolerance Is Associated with Melatonin Synthesis and Metabolism in Maize

**DOI:** 10.3390/ijms21030782

**Published:** 2020-01-25

**Authors:** Luying Sun, Fengbin Song, Junhong Guo, Xiancan Zhu, Shengqun Liu, Fulai Liu, Xiangnan Li

**Affiliations:** 1Key Laboratory of Mollisols Agroecology, Northeast Institute of Geography and Agroecology, Chinese Academy of Sciences, Changchun 130102, China; sunluy1@126.com (L.S.); guojunhongiga@163.com (J.G.); lsq@iga.ac.cn (S.L.); 2University of Chinese Academy of Sciences, Beijing 100049, China; 3College of Life Sciences, Anhui Normal University, Wuhu 241000, China; zhuxiancan@iga.ac.cn; 4Department of Plant and Environmental Sciences, University of Copenhagen, Taastrup 2630, Denmark; fl@plen.ku.dk

**Keywords:** tryptophan, nanoparticle, serotonin, mitochondria, chloroplast

## Abstract

The applications of ZnO nanoparticles in agriculture have largely contributed to crop growth regulation, quality enhancement, and induction of stress tolerance, while the underlying mechanisms remain elusive. Herein, the involvement of melatonin synthesis and metabolism in the process of nano-ZnO induced drought tolerance was investigated in maize. Drought stress resulted in the changes of subcellular ultrastructure, the accumulation of malondialdehyde and osmolytes in leaf. The nano-ZnO (100 mg L^−1^) application promoted the melatonin synthesis and activated the antioxidant enzyme system, which alleviated drought-induced damage to mitochondria and chloroplast. These changes were associated with upregulation of the relative transcript abundance of *Fe/Mn SOD*, *Cu/Zn SOD*, *APX*, *CAT*, *TDC*, *SNAT*, *COMT*, and *ASMT* induced by nano-ZnO application. It was suggested that modifications in endogenous melatonin synthesis were involved in the nano-ZnO induced drought tolerance in maize.

## 1. Introduction

Drought is one of the major abiotic stresses that largely limit crop production and threaten global food security. Drought influences plant growth and development at various levels [1]. For instance, at a sub-cellular level, drought-induced lipid peroxidation causes reorganization of the chloroplast ultrastructure [2]. Additionally, drought damages the stroma lamellae and grana of the chloroplast and decreases the accumulation of starch granules [3]. Thus, protecting the cell ultrastructure under drought is of primarily importance for enhancing plant performance under drought.

ZnO nanoparticles (nano-ZnO) have been widely applied due to their unique physiochemical properties and large specific surface area [4,5]. Nanomaterials regulate plant growth and development at multiple levels [6]. For instance, the beneficial effect of nano-ZnO at low concentrations has been well documented in peanut [7], mung bean [8], and tomato [9], suggesting that nanoparticles induce the tolerance to various abiotic stresses, including drought [10], salt [11], and low temperature [12]. The alleviation of abiotic stress by nanoparticles was often associated with the enhanced antioxidant enzymes activities [13]. In addition, modulation of phytohormones levels was involved in the nanoparticles-induced stress tolerance [14]. For example, nano-Fe_2_O_3_ (1000 mg kg^−1^) application increases the concentrations of gibberellic acid (GA) and zeatin riboside (ZR), which benefits the growth of peanut [15]. The nanomaterials (nano-Fe_2_O_3_ or nano-TiO_2_) suspension (50 mg L^−1^) maintains the integrity of the cell structure and suppresses turnip mosaic virus infection by altering the concentrations of ZR, brassinolide (BR), and abscisic acid (ABA) in tobacco [16]. In addition, nano-Fe_2_O_3_ (80 mg L^−1^) treatment increases the concentrations of jasmonic acid and 12-oxo phytodienoic acid, thereby enhancing the stress tolerance in watermelon [17].

Melatonin (*N*-acetyl-5-methoxy-tryptamine) is an effective free radical scavenger and naturally occurring antioxidant in response to abnormal cellular metabolic and abiotic stress [18]. The original sites of melatonin biosynthesis are mitochondria and chloroplasts in higher plants [19,20,21]. Normally, melatonin biosynthesis is based on tryptophan, which is catalyzed by tryptophan decarboxylase (TDC) and tryptamine 5-hydroxylase (T5H) and converted into serotonin. Then serotonin is converted into melatonin by serotonin *N*-acetyltransferase (SNAT) and *N*-acetylserotonin *O*-methyltransferase (ASMT)/caffeic acid *O*-methyltransferase (COMT) [22]. Under stress conditions, the endogenous melatonin level normally increased through alterations in biosynthesis and metabolism pathways, to protect mitochondria and chloroplast against oxidative stress [23]. Once synthesized, melatonin is rapidly metabolized into 2-hydroxymelatonin (2-HM). Byeon and Back [24] observed that the accumulation of 2-hydroxymelatonin was higher than melatonin. The melatonin metabolites, such as 2-HM and *N*^1^-acetyl-*N*^2^-formyl-5-methoxyknuramine (AFMK), have been proven to have higher antioxidant effects [25]. Previous studies found that not only melatonin but also its precursors and metabolites play crucial roles in stress response in plants [26]. It was found that the accumulation of serotonin attributes to the scavenging of ROS, hence, delaying senescence in rice leaves [27]. Lee and Back [28] documented 2-hydroxymelatonin promotes the tolerance to various abiotic stresses, partly through induction of the *Myb4* TF in rice. In addition, the reduction of melatonin synthesis caused by the suppression of the *SNAT* and *ASMT* leads to high sensitivity to abiotic stress [24].

Currently, the applications of nanomaterials lead to great breakthroughs in the regulation of plant growth and the alleviation of stresses in agriculture. The zinc is a necessary micronutrient in regulating various metabolic pathways in plants [29]. The grain yield in nano-ZnO-treated (400 mg L^−1^) maize was 42% and 15% higher than that in the control and ZnSO_4_ treated maize [30]. However, the physiochemical mechanisms of nano-ZnO-induced drought tolerance in maize remain largely unknown. Here, we hypothesized that nano-ZnO application could modulate the melatonin synthesis and metabolism contributing to drought tolerance in maize plants.

## 2. Results

### 2.1. Characteristics of Nano-ZnO

The phase purity of nano-ZnO particles used in this study was verified by the X-ray diffractograms (XRD) (Figure 1A). The absence of additional impurity peaks indicated nano-ZnO phase purity. Using TEM, the nano-ZnO was detected to be nearly spheroidal to oblong in shape, and the average size of the particles was 37.7 ± 15.5 nm (Figure 1B). The 100 mg L^−1^ nano-ZnO suspension showed an absorption peak at 376 nm (Figure 1C). The hydrodynamic size of nano-ZnO was around 300 nm, which indicated the presence of aggregates and/or agglomerates of nano-ZnO in suspension (Figure 1D). For zeta potential distribution, the 100 mg L^−1^ nano-ZnO denoted monodispersity with one peak at 14 mV (Figure 1E).

### 2.2. Chloroplast and Mitochondria of Maize Plants

In the non-stress control, the chloroplasts in maize leaf were located near the cell wall and had an uninjured outer membrane with clear boundaries (Figure 2). Chloroplast grana and stroma lamella were arranged parallel to the long axis of the chloroplast. There were small high-electron density osmium granules, and the size of the starch granules was relatively small in chloroplasts. The mitochondria were abundant, which were regular spheroid or ellipsoid in the cytoplasm. Its structure was complete, and the cristae was clearly visible. Compared with non-ZnO treatment (N) plants, there was no obvious alteration in the ultrastructure of leaf mesophyll cells in the nano-ZnO-treated plants under non-stress condition.

Under drought, chloroplast ultrastructure was reorganized, including distortion of chloroplast morphology, changes of membrane system, and increase of osmium particle. The starch grains were degraded, while vacuoles appeared in the chloroplast lamella. In addition, most of the mitochondria disintegrated under drought stress. It should be noted that the ultrastructure of mesophyll cells in nano-ZnO plants showed no obvious changes under drought.

### 2.3. Soluble Sugar, Proline, and Soluble Protein

Drought (D) significantly increased the proline concentration, while decreased the soluble protein concentration, compared with the non-stress control (W) (Figure 3; Table 1). However, drought had no significant effect on soluble sugar concentration in leaf in both nano-ZnO and N plants. Compared with N plants, the concentrations of soluble sugar and soluble protein in nano-ZnO plants were significantly higher under two water regimes. Meanwhile, two-way ANOVA output indicated the significant interactive effects of nano-ZnO × water regime on proline concentration. Compared with N plants, the nano-ZnO plants had higher proline concentration in leaf under drought, while a reversed trend was found under non-stress condition.

### 2.4. Antioxidant Enzyme Activity

Significantly higher concentrations of malondialdehyde (MDA) and H_2_O_2_ and activities of total superoxide dismutase (SOD) and ascorbate peroxidase (APX) in maize leaf were found under drought, in relation to the non-stress control (Table 1, Figure 4). The SOD and catalase (CAT) activities were significantly increased by nano-ZnO plants, compared with N plants under both non-stress and drought conditions. The H_2_O_2_ concentration was significantly reduced in nano-ZnO plants, compared with N plants under both conditions. Interestingly, the trend of APX activities in nano-ZnO plants were like that in N plants under non-stress condition. Under drought, the APX activity in nano-ZnO plants was significantly higher than that of N plants. In addition, two-way ANOVA indicated the significant interactive effect of nano-ZnO × water regime on APX activity.

### 2.5. Melatonin Metabolism

Drought decreased the tryptamine concentration significantly in leaves; however, there was no significant effect on tryptophan, serotonin, *N*-acetylserotonin (NAS), and melatonin concentrations, compared with the non-stress control (Table 1). Under drought, the tryptophan, tryptamine, serotonin, and melatonin concentrations were significantly increased by nano-ZnO treatment, compared with N treatment (Figure 5). It should be noted that the serotonin concentration in nano-ZnO plants was 10.8% and 83.5% higher than that of N plants under non-stress and drought. The melatonin concentration in nano-ZnO plants was 22.8% and 77.8% higher than that in N plants under non-stress and drought. In addition, two-way ANOVA showed a significant interactive effect of nano-ZnO × water regime on tryptamine concentration in maize leaves.

Significantly higher 2-HM and AFMK concentrations were found in drought treatments, compared with the control (Table 1, Figure 6). Nano-ZnO treatment significantly increased 2-HM and AFMK concentrations under non-stress condition, while it significantly reduced the levels of 2-HM and AFMK under drought. The outputs of two-way ANOVA showed a significant effect of interaction of nano-ZnO × water regime on 2-HM and AFMK concentrations.

### 2.6. Gene Expression

The relative transcript abundance of *Cu/Zn SOD*, *Fe/Mn SOD*, and *APX* was significantly up-regulated by drought (Figure 7). The relative transcript abundance of *Fe/Mn SOD*, *Cu/Zn SOD*, *APX*, and *CAT* in nano-ZnO plants was 15.2%, 54.8%, 7.4%, and 19.0% higher than that of N plants under drought. Significant interactive effects of nano-ZnO × water regime on the expressions of *Cu/Zn SOD*, *Fe/Mn SOD*, and *CAT* were found (Table 2).

Compared with the non-stress control, the relative transcript abundances of *TDC* and *SNAT* were significantly down-regulated by drought (Figure 8 and Figure 9). The relative transcript abundance of *TDC*, *ASMT*, *COMT*, and *SNAT* in nano-ZnO plants was 369.6%, 107.0%, 40.4%, and 123.2% higher than that in N plants under drought, respectively. In addition, the significant effect of interaction of nano-ZnO × water regime on the expressions of *TDC*, *ASMT*, and *COMT* was observed (Table 2).

### 2.7. Correlations

A significant positively linear relationship was found between tryptophan concentration and CAT activity in maize leaves (Table 3). Tryptamine concentration was significantly positively correlated to soluble protein concentration, while negatively correlated to the concentrations of proline, H_2_O_2_, and MDA and the activity of APX, respectively. Serotonin concentration was significantly positively correlated to concentrations of melatonin, soluble sugar, and soluble protein, respectively. In addition, the proline concentration was positively correlated to the NAS concentration in leaves.

## 3. Discussion

Mitochondria possess a remarkable flexibility in energy dissipation and electron transfer, both are important for drought tolerance in plants [31]. The respiration in mitochondria alleviates the effects of drought on the intracellular environment in plants [32]. In the present study, the chloroplast and mitochondria in N plants were adversely affected by drought, by resulting in the destruction of membrane and lamellar structure formation in chloroplast [33]. Nano-ZnO increased the structure stabilization of chloroplast and mitochondria under drought, which helped to maintain a better photosynthesis. Positive effects on chloroplast and mitochondria have been reported for various metal oxide nanoparticles. For instance, nano-TiO_2_ protects the chloroplast and mitochondria from environmental stress via increasing antioxidant enzyme activities and up-regulating *At3g23990* [34,35]. Similarly, the single-walled carbon nanotubes increase the photosynthetic activity and the threshold of electron transport due to their protective effects on lipid membranes in chloroplast [36]. In addition, Sharif et al. [37] documented that the accumulation of endogenous melatonin can repair mitochondria damaged by drought stress and enhances nitric oxide concentration and antioxidant enzymes activities, hence, improving plant drought tolerance. Thus, the nano-ZnO application might cause the alteration of membrane potential and fluidity of the endomembrane system and stimulate the antioxidant defense system to protect the structural and functional integrity of the membrane system, which contributes to drought tolerance.

Osmotic adjustment through accumulating osmolytes maintains higher cell turgor, hence, protecting plants from drought [38]. Higher proline and soluble sugar concentrations caused by nano-ZnO treatment are related to the salt tolerance in maize [39]. In this study, the soluble protein, sugar, and proline concentrations were all higher in nano-ZnO plants than N plants under drought, indicating that nano-ZnO enhanced the osmotic regulation. The endogenous proline level depends on its synthesis and degradation, which were catalyzed by Δ1-pyrroline-5-carboxylate synthetase (P5CS), ornithine aminotransferase (OAT), and proline dehydrogenase (PDH) [40]. The increment of proline caused by enhancement of P5CS and OAT activities and the reduction of PDH activity under cold stress is well documented [41]. Soluble protein has a positive effect on cell membrane [42]. Consistent with this, in cluster bean, it was also found that nano-ZnO treatment increased the total soluble protein concentration [43].

Drought results in accumulation of ROS, leading to lipid peroxidation of the cell membrane system, which is mainly due to the production of MDA. Herein, nano-ZnO significantly increased the activities of SOD, CAT, and APX, which reduced the accumulation of H_2_O_2_ under drought. In agreement with this, the relative transcript abundance of *Fe/Mn SOD*, *Cu/Zn SOD*, *APX*, and *CAT* were significantly up-regulated in nano-ZnO plants under drought, as compared with the N plants. Similarly, nano-ZnO minimizes the oxidative effect of drought in wheat [44]. Thus, the nanoparticles were likely involved in scavenging ROS by activating antioxidant enzyme systems [45]. Significantly higher antioxidant enzymes activities in nano-ZnO plants reduced the H_2_O_2_ and MDA levels in maize leaves, which reduce the drought-induced lipid peroxidation, hence maintaining the stabilization of chloroplasts and mitochondria [46].

Numerous studies have indicated melatonin, as an effective antioxidant and/or regulator of plant growth and development, is involved in the induction of drought tolerance [47,48]. Further, the precursors of melatonin synthesis were related to the stress tolerance [49]. In the present study, the results documented that nano-ZnO significantly enhanced the metabolic pathways of tryptophan in maize under drought. As the precursor of auxins and melatonin, higher tryptophan and tryptamine concentrations might contribute to drought tolerance via alleviating oxidative damage and regulating osmotic balance [50]. The initial precursor of melatonin synthesis is tryptophan, which is catalyzed by TDC and T5H to synthesize the intermediate of melatonin, i.e., tryptamine and serotonin [25]. In addition, the L-tryptophan is the precursor for auxins production, which benefits for the osmotic balance and plant growth through regulating the osmotic pressure of vacuolar and mediating the translocation of metabolites required by cell elongation under drought [51]. Meanwhile, tryptamine is a precursor of indole-3-acetic acid [52], which is related to the drought tolerance in plants [53]. In addition, nano-ZnO treatment caused a significant boost in serotonin and melatonin levels in maize leaves under drought. The process of serotonin synthesis melatonin is catalyzed by SNAT and ASMT/COMT. In addition to regulating plant growth and morphogenesis, both melatonin and serotonin play an important role in the defense against abiotic stress. For instance, serotonin improves salt tolerance through mediating the influx of ions into chloroplast [54]. Evidence has indicated that exogenous melatonin application induced a significant increase in endogenous melatonin concentration, hence, promoting drought tolerance through increasing the antioxidant enzymes activities by up-regulating the expressions of *SOD*, *CAT*, and *POD* in maize [55]. Higher melatonin concentration enhanced drought tolerance by regulating nitro-oxidative balance and proline metabolism, hence, improving biomass production and photoprotection [55]. In this study, the NAS concentration was neither affected by drought nor nano-ZnO treatment. Although NAS exhibits antioxidant ability in animals, the role of NAS in plant against abiotic stress has not been evidenced [56]. Melatonin is decomposed by enzymatic reaction, pseudo-enzyme reaction, and cascade interaction with free radicals to produce a variety of metabolites, including 2-hydroxymelatonin and AFMK. Galano and Reiter [57] documented that AFMK might inhibit OH production via Fenton-like reactions. Melatonin is catalyzed to produce 2-HM by multiple M2H enzymes, which are classified in the 2-oxoglutarate-dependent dioxygenases (2-ODD) superfamily. Genes encoding 2-ODD members are involved in the synthesis of flavonoid. 2-HM, whose biological activity is half of that of melatonin, is a signaling molecule inducing the expression of defense related genes [58]. Lee and Back [59] documented that 2-HM-induced MAP kinase activation was similar to that of melatonin in *Arabidopsis*. Here, it was found that 2-HM and AFMK concentrations were unaffected by nano-ZnO and drought, suggesting that 2-HM and AFMK were not involved in the nano-ZnO-induced drought tolerance in maize. Under drought, nano-ZnO increased the melatonin concentration mainly through enhancing the upstream substrate concentrations and gene expression of key enzymes in melatonin biosynthesis.

As a ROS scavenger, melatonin plays key roles in drought response in plants. The endogenous melatonin level depends on the balance of melatonin synthesis and metabolism. TDC is identified as the first rate-limiting enzyme in the melatonin synthesis pathway. Overexpression of *TDC* increases melatonin and melatonin intermediate levels [60]. SNAT and ASMT are also two key enzymes involving in the melatonin synthesis [61]. Conversely, it has been documented that the over-expression of *SNAT* in chloroplasts and cytoplasm increased the SNAT enzyme activity, while had no effect on melatonin concentration. This might be due to that the melatonin synthesis is multiple factors dependent [62]. In addition, serotonin can be also catalyzed by COMT to produce 5-methoxytryptamine, which is further converted into melatonin. The overexpression of *TaCOMT* enhances drought resistance in transgenic *Arabidopsis* via increasing the melatonin and proline levels and decreasing MDA concentration [63]. Further, COMT contributed to stem mechanical strength by enhancing lignin synthesis, which also benefited the drought tolerance in maize [64]. Previous studies reported that transgenic tomato overexpressing ovine *ASMT* had higher melatonin production and enhanced drought tolerance [65]. In maize, the overexpression of *MzASMT1* also increased the melatonin concentration, hence, promoting the drought tolerance [66]. Consistent with many studies, in this study the nano-ZnO led to up-regulated expressions of *TDC*, *SNAT*, *COMT*, and *ASMT* to enhance melatonin synthesis under drought, compared with the non-nano treatment.

In conclusion, drought caused the changes in the ultrastructure of chloroplast and mitochondria and the accumulation of MDA, H_2_O_2_, and osmolytes in maize. The nano-ZnO (100 mg L^−1^) up-regulated the melatonin synthesis pathways and activated the antioxidant enzyme system, which benefited the alleviation of drought-induced damage to mitochondria and chloroplast, hence, enhancing the drought tolerance. Consistent with these modifications, the relative transcript abundances of *Fe/Mn SOD*, *Cu/Zn SOD*, *APX*, *CAT*, *TDC*, *SNAT*, *COMT*, and *ASMT* were all up-regulated by nano-ZnO application. It is suggested that the nano-ZnO application-induced drought tolerance involves the up-regulation of endogenous melatonin synthesis in maize.

## 4. Materials and Methods

### 4.1. Nanoparticles

Nano-ZnO (code XFZn01) used in this study was produced by XFNANO Materials Tech Co. (Nanjing, China). The particles were white single crystal, 99.9% purity, 20 nm in average size, and 21.5 m^2^ g^−1^ of specific surface. Suspensions of nano-ZnO were prepared at 100 mg L^−1^ (particle concentration) in H_2_O, stirred for 30 min and then sonicated for 30 min. The morphology and size of nano-ZnO suspension were characterized in the Results section.

### 4.2. Experimental Setup

Four maize (*Zea mays* L. cv. Jidan 27) seeds were sown in plastic pots containing 1.1 kg soil. Two seedlings were retained after thinning, and the suspensions of nano-ZnO were added into half of these pots at 7 days after emergence. A total amount of 1000 mL nano-ZnO suspensions (100 mg L^−1^) was added in thrice. Twenty days after nano-ZnO treatment, half of the nano-ZnO-treated and the control plants were exposed to a 6-day drought stress treatment by withholding watering until the relative soil water content reached approximately 45%. Thus, four treatments were established: ZnO-W, nano-ZnO treatment, and normal water supply; N-W, non-ZnO treatment, and normal water supply; ZnO-D, nano-ZnO treatment, and drought stress; N-D, non-ZnO treatment, and drought stress. A randomized block design was used in the experiment with three replicates for each treatment. The mixture of samples from four plants were used as a replicate. All pots were placed randomly in a climate chamber at 25/20 °C (day/night) with 75%–85% relative humidity and 14 h of photoperiod.

### 4.3. Physicochemical Characterization of Nano-ZnO

The phase compositions and average crystallite sizes were conducted by X-ray diffractograms (XRD). The morphology and structure were determined by transmission electron microscopy (TEM). The hydrodynamic size distribution and zeta potential of nano-ZnO (100 mg L^−1^) suspended in distilled and deionized water were measured by dynamic light scattering (DLS) and zeta potential. In addition, the UV-visible spectra were measured by an Ultra-visible spectrophotometer [67].

### 4.4. Transmission Electron Microscope (TEM) Analysis

Samples (1 mm × 1 mm) were taken from near the center vein of each leaf (applied in 2.5% glutaraldehyde on ice bed). The samples were fixed with 2.5% glutaraldehyde (pre-cooling), then pumped with a vacuum pump until the leaf materials floating in glutaraldehyde sunk to the bottom, and stored in 4 °C refrigerator for measurements. For the transmission electron microscope analysis (Hitachi HT7700, Tokyo, Japan), the samples were washed, dehydrated, and embedded with paraffin. Thin slices were removed from the samples with an ultra-micro diamond knife (Leica EM UC6) and stained [2].

### 4.5. Soluble Sugar, Proline, and Soluble Protein Concentrations

Anthrone method was applied to the determination of soluble sugar concentration [68]. Proline concentration was measured following the indene triketone method [39]. Coomassie brilliant blue staining method was used to detect soluble protein concentration [68].

### 4.6. MDA and H_2_O_2_ Concentrations and Antioxidant Enzyme Activity

Malondialdehyde (MDA) concentration was measured according to Anjum et al. [46]. Total superoxide dismutase (SOD) activity was measured by the inhibition of photochemical reduction of nitroblue tetrazolium (NBT) [69]. Catalase (CAT) activity was measured according to Li et al. [65]. Ascorbate peroxidase (APX) activity was measured by monitoring the decrease at 290 nm according to the protocol of Zheng et al. [70] with a few modifications. The reaction system contained 0.1 mL enzyme extraction, 2.75 mL buffer (50 mM HEPES + 300 μM AsA + 100 μM EDTA Na_2_, pH 7.0), and 0.15 mL 30% H_2_O_2_. Leaf H_2_O_2_ concentration was determined by monitoring the absorbance of titanium peroxide complex at 410 nm according to Zheng et al. [70].

### 4.7. High-Performance Liquid Chromatography (HPLC) Analysis of Tryptophan, Tryptamine, and Serotonin

Leaf samples (0.5 g) were mixed with cooled 5% perchloric acid for 300 s, and ultrasonically extracted in a cold water bath for 1 h. The homogenates were centrifuged at 5000 *g* for 10 min at 4 °C. Five mL supernatant with 1 mL NaOH and 20 μL benzoyl chloride were vortexed for 30 s and reacted in water bath for 20 min, during which vortexing was performed for 30 s every 5 min. Then 2 mL saturated NaCl and 2 mL anhydrous ether were added, shaken, and blown dry with nitrogen, which was dissolved with 1 mL methanol and filtered with membrane filter (0.22 μm). The supernatants were analyzed by HPLC system. Samples were separated on an Atlantis C18 column (4.6 mm × 250 mm) with mobile phase containing 5% (*v*/*v*) methanol, 30% (*v*/*v*) acetonitrile, and 65% (*v*/*v*) ammonium acetate at a flow rate of 1.0 mL min^−1^. Detection of samples were monitored at 254 nm. All measurements were reproduced in triplicate [71].

### 4.8. High-Performance Liquid Chromatography (HPLC) Analysis of Melatonin

Melatonin concentration was measured according to Byeon et al. [59] with a few modifications. Leaf samples (0.2 g) were pulverized to powder in liquid nitrogen and extracted with 10 mL chloroform for 1 h and ultrasound for 15 min. Then the extracts were centrifuged at 10,000× *g* for 15 min at 4 °C and evaporated with nitrogen gas to dry. The substance was dissolved in 1 mL 80% methanol and filtered by a membrane filter (0.22 µm). 10 µL samples were analyzed by HPLC system with a fluorescence detector system (Waters, Milford, MA, USA). All measurements were determined in triplicate.

### 4.9. Quantification of N-Acetylserotonin (NAS), 2-Hydroxymelatonin (2-HM), and N^1^-Acetyl-N^2^-Formyl-5-Methoxyknuramine (AFMK)

Leaf samples (0.2 g) were ground into a fine powder with liquid nitrogen and extracted with 1.8 mL 0.1 M extraction mixture (acetone: methanol: water (*v*/*v*/*v*) = 89:10:1) in a 2.0 mL centrifuge tube. The mixture was centrifuged at 10,000× *g* for 15 min at 4 °C and supernatant was collected for measurement. The *N*-acetylserotonin (NAS), 2-hydroxymelatonin (2-HM), and *N*^1^-acetyl-*N*^2^-formyl-5-methoxyknuramine (AFMK) were detected using enzyme-linked immunosorbent assay (ELISA) Kit (Shanghai Youxuan Biotech Co., Ltd., Shanghai, China), and their concentrations were measured by Epoch Microplate Spectrophotometer (BioTek Instruments, Inc., Winosky, VT, USA).

### 4.10. RNA Extraction and Quantitative Real-Time Polymerase Chain Reactions (RT-qPCR)

Total RNA was extracted using Trizol (Sangon Biotech, Shanghai, China) and cDNA was reverse-transcribed using PrimeScript^TM^ RT reagent Kit with gDNA Eraser (TaKaRa, Shiga, Japan) according to the manufacturer’s instructions, respectively. The RT-qPCR was carried out using the PowerUp^TM^ SYBR^TM^ Green Master Mix (Thermo Fisher Scientific, Inc., Waltham, MA, USA) on QuantStudio 1 Real-Time PCR System (Thermo Fisher Scientific, Inc., Waltham, MA, USA). The reaction system: 10 µL 2 × PowerUp^TM^ SYBR^TM^ Green Master Mix, 1 µL cDNA solution, 0.4 µL forward primers (10 µM), 0.4 µL reverse primers (10 µM), 8.2 µL ddH_2_O. The PCR program was as follows: 50 °C for 2 min, 95 °C for 2 min, 40 cycles at 95 °C for 15 s, 60 °C for 15 s, 72 °C for 30 s. Melting curves were used to verify the amplification specificity via a stepwise heating of the amplicon from 60 to 95 °C at a ramp rate of 0.15 °C s^−1^. In this study, *Zm**Actin* was used as the internal gene. Gene-specific primers are listed in Table 4. Each extraction and RT-qPCR were replicated three times. The relative expression levels of the target gene were calculated by the 2^−^^△^^△Ct^ method.

### 4.11. Data Analysis

Statistical analysis was performed using Statistical Package for the Social Sciences (SPSS 22.0, SPSS Inc., Chicago, IL, USA). All data were first tested for homogeneity of variance and then tested by two-way ANOVA.

## Figures and Tables

**Figure 1 ijms-21-00782-f001:**
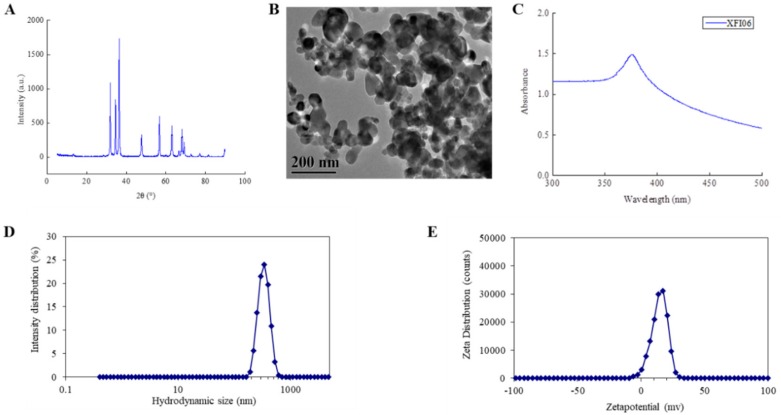
(**A**) X-ray diffractograms (XRD) pattern of nano-ZnO; (**B**) the transmission electron microscopy (TEM) image of nano-ZnO (code XFI06); (**C**) the UV-visible spectra of 100 mg L^−1^ nano-ZnO; (**D**) the hydrodynamic size distribution of nano-ZnO; (**E**) the zeta potential distribution of nano-ZnO.

**Figure 2 ijms-21-00782-f002:**
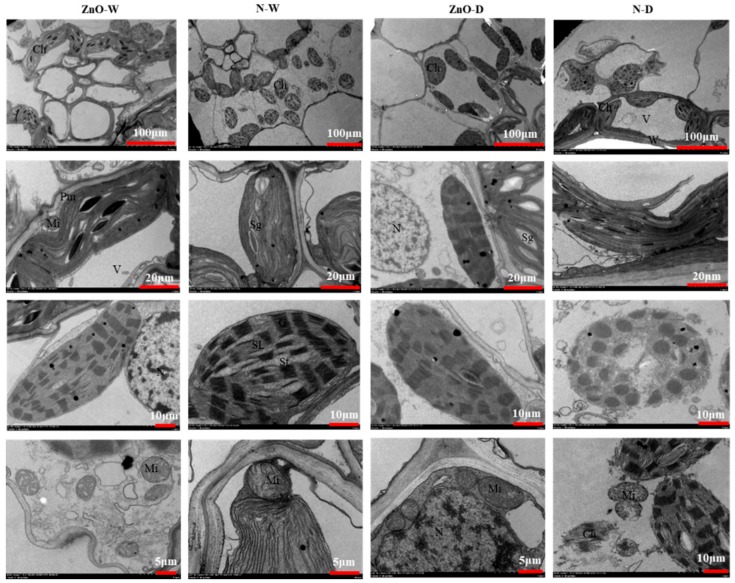
TEM images of chloroplast and mitochondria in maize leaves as affected by nano-ZnO and drought stress. ZnO, nano-ZnO treatment; N, non-ZnO treatment; W, normal water supply; D, drought stress. Ch, chloroplast; Mi, mitochondrion; G, granum; SL, stoma lamellae; St, stroma; N, nucleus; V, vacuolar; Pm, plasma membrane; W, cell wall; Sg, starch granule.

**Figure 3 ijms-21-00782-f003:**
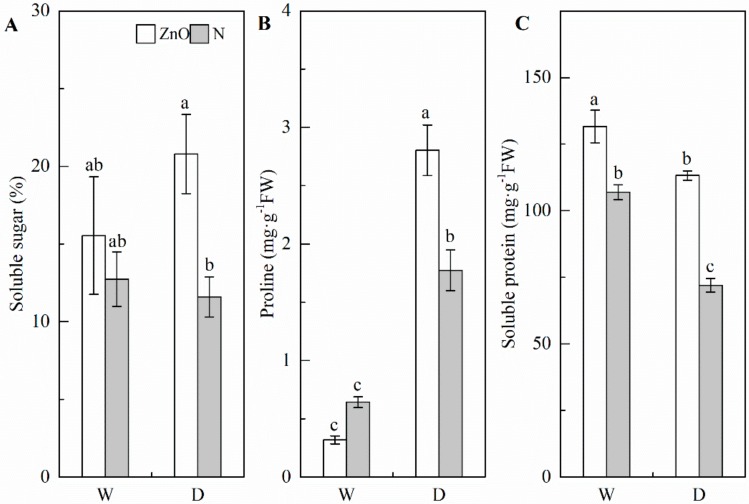
Concentrations of soluble sugar (**A**), proline (**B**), and soluble protein (**C**) in maize leaves as affected by nano-ZnO and drought stress. Data are reported as mean ± SE (*n* = 3). Different small letters indicate significant difference at *p* < 0.05 level. ZnO, nano-ZnO treatment; N, non-ZnO treatment; W, normal water supply; D, drought stress.

**Figure 4 ijms-21-00782-f004:**
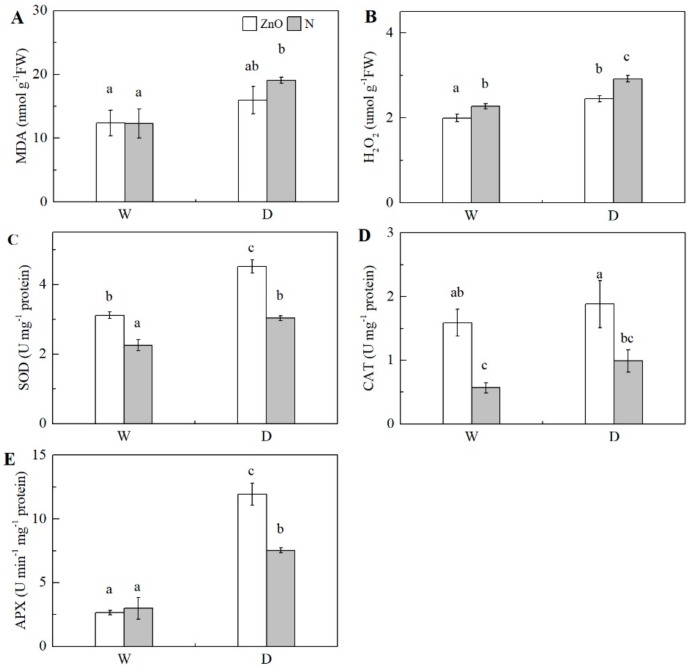
Concentrations of MDA and H_2_O_2_ (**A**, **B**) and activities of SOD (**C**), CAT (**D**), and APX (**E**) in maize leaves as affected by nano-ZnO and drought stress. Data are reported as mean ± SE (*n* = 3). Different small letters indicate significant difference at *p* < 0.05 level. ZnO, nano-ZnO treatment; N, non-ZnO treatment; W, normal water supply; D, drought stress.

**Figure 5 ijms-21-00782-f005:**
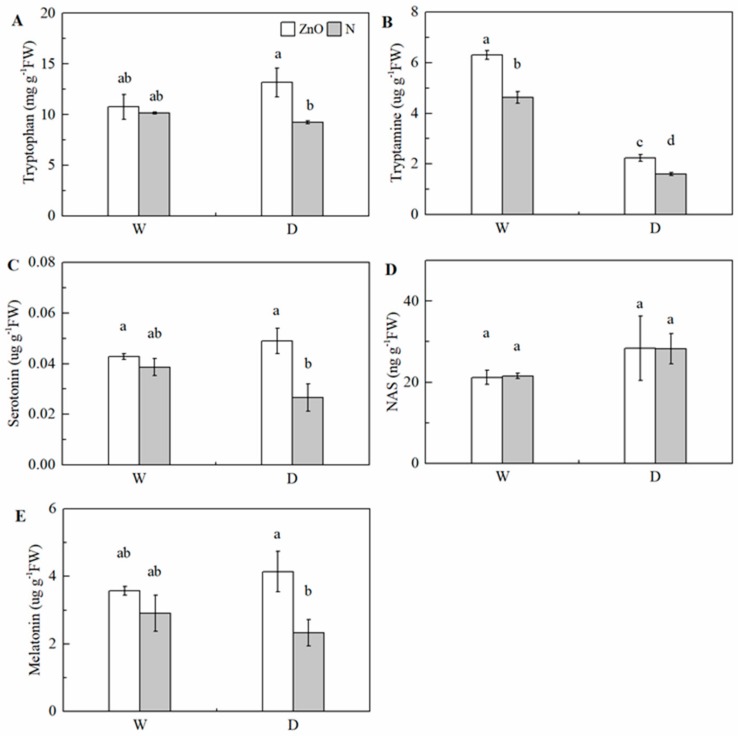
Concentrations of tryptophan (**A**), tryptamine (**B**), serotonin (**C**), NAS (**D**), and melatonin (**E**) in maize leaves as affected by nano-ZnO and drought stress. Data are reported as mean ± SE (*n* = 3). Different small letters indicate significant difference at *p* < 0.05 level. NAS, *N*-acetylserotonin; ZnO, nano-ZnO treatment; N, non-ZnO treatment; W, normal water supply; D, drought stress.

**Figure 6 ijms-21-00782-f006:**
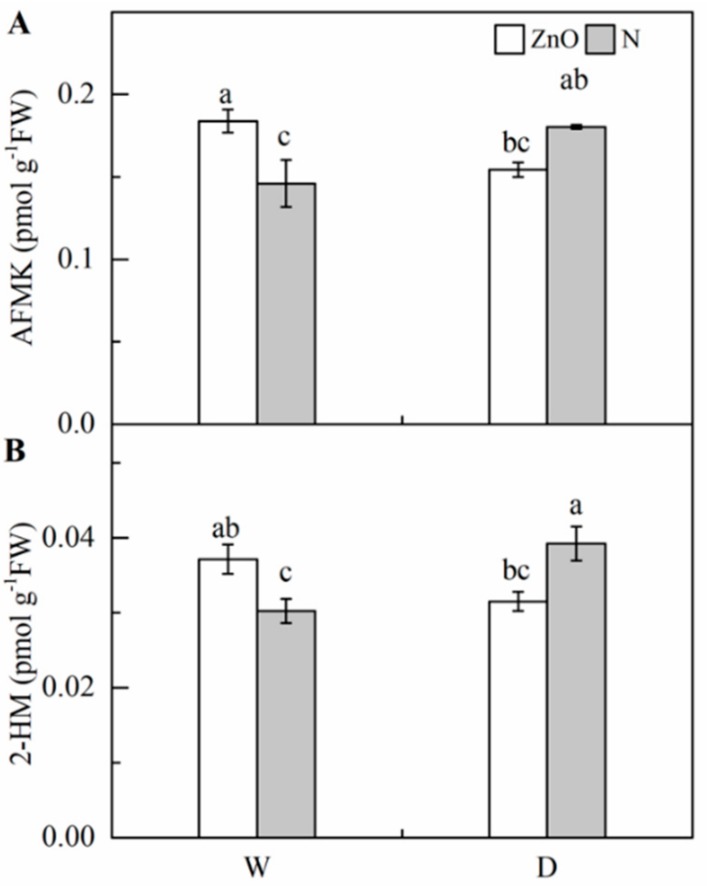
Concentrations of AFMK (**A**) and 2-HM (**B**) in maize leaves as affected by nano-ZnO and drought stress. Data are reported as mean ± SE (*n* = 3). Different small letters indicate significant difference at *p* < 0.05 level. 2-HM, 2-hydroxymelatonin; AFMK, *N*^1^-acetyl-*N*^2^-formyl-5-methoxyknuramine; ZnO, nano-ZnO treatment; N, non-ZnO treatment; W, normal water supply; D, drought stress.

**Figure 7 ijms-21-00782-f007:**
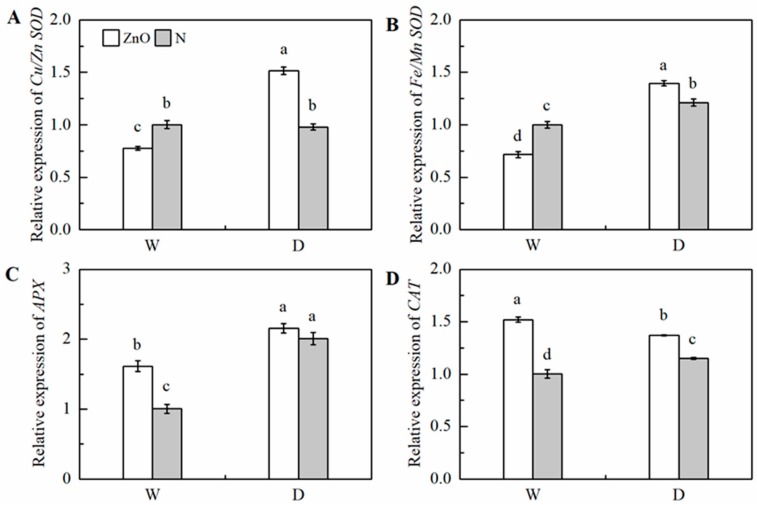
The relative transcript abundance of *Cu/Zn SOD* (**A**)*, Fe/Mn SOD* (**B**), *APX* (**C**), and *CAT* (**D**) in maize leaves as affected by nano-ZnO and drought stress. Data are reported as mean ± SE (*n* = 3). Different small letters indicate significant difference at *p* < 0.05 level. ZnO, nano-ZnO treatment; N, non-ZnO treatment; W, normal water supply; D, drought stress.

**Figure 8 ijms-21-00782-f008:**
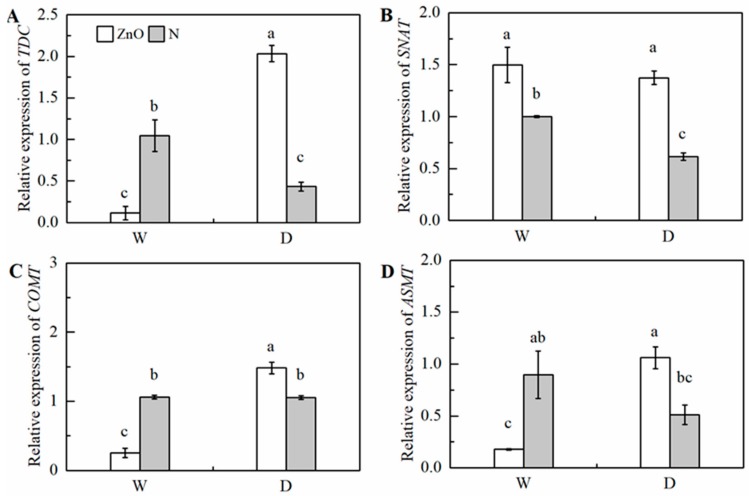
The relative transcript abundance of *TDC* (**A**)*, SNAT* (**B**), *COMT* (**C**), and *ASMT* (**D**) in maize leaves as affected by nano-ZnO and drought stress. Data are reported as mean ± SE (*n* = 3). Different small letters indicate significant difference at *p* < 0.05 level. ZnO, nano-ZnO treatment; N, non-ZnO treatment; W, normal water supply; D, drought stress.

**Figure 9 ijms-21-00782-f009:**
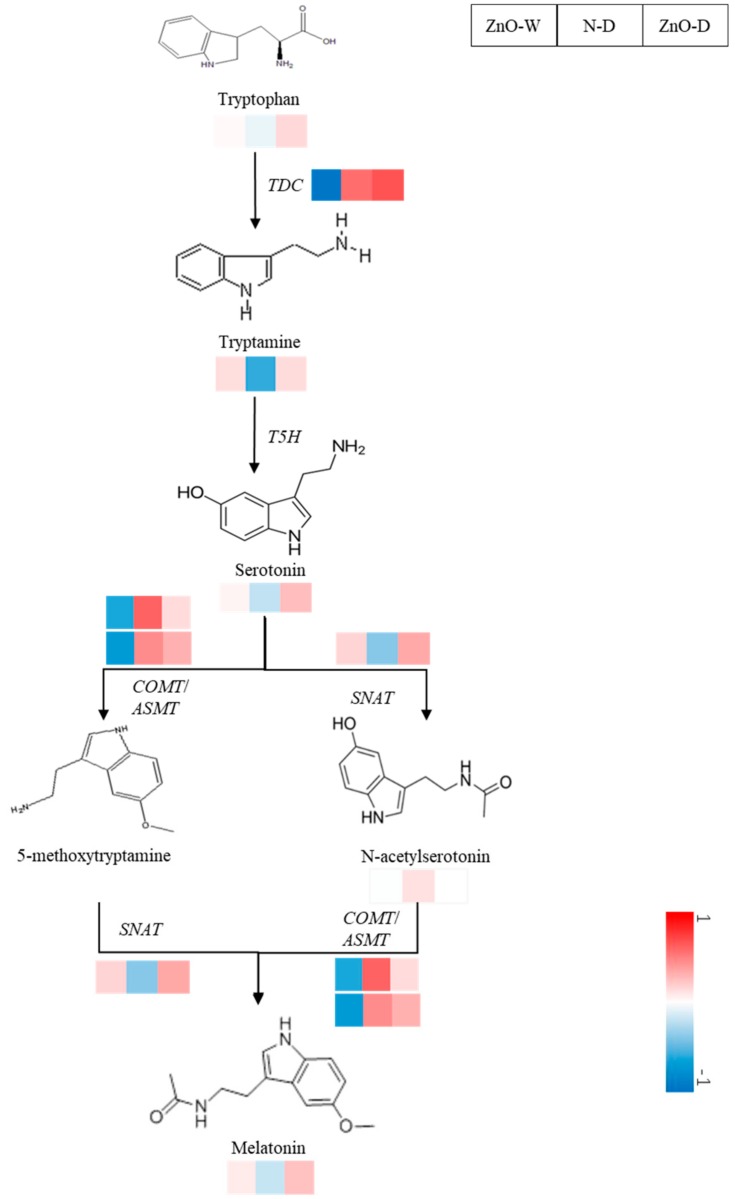
Modifications of relative expression of related genes and concentrations of metabolites involved in melatonin synthesis in maize leaf induced by nano-ZnO and drought stress. Compared with N-W, the differences in genes expression and intermediate concentrations level for a given enzyme under ZnO-W, N-D and ZnO-D, were log-normalized and converted to a color scale. It was reorganized after analysis with the PageMan software. Up-regulation and down-regulation were indicated in increasing red and blue, respectively. The unaltered level was indicated in white. TDC, tryptophan decarboxylase; ASMT, *N*-acetylserotonin *O*-methyltransferase; COMT, caffeic acid *O*-methyltransferase; SNAT, serotonin *N*-acetyltransferase; ZnO, nano-ZnO treatment; N, non-ZnO treatment; W, normal water supply; D, drought stress.

**Table 1 ijms-21-00782-t001:** Output of two-way ANOVA for the interactive effects of drought stress and nano-ZnO on the tested parameters in maize.

	Nano-ZnO	Water Regime	Nano-ZnO × Water Regime
Soluble sugar	*	ns	ns
Proline	**	**	**
Soluble protein	**	**	ns
MDA	ns	*	ns
SOD	**	**	ns
APX	*	**	**
CAT	**	ns	ns
H_2_O_2_	**	**	ns
Tryptophan	*	ns	ns
Tryptamine	**	**	*
Serotonin	*	ns	ns
NAS	ns	ns	ns
Melatonin	*	ns	ns
2-HM	ns	ns	**
AFMK	ns	ns	*

MDA, malondialdehyde; SOD, total superoxide dismutase; APX, ascorbate peroxidase; CAT, catalase; H_2_O_2_, hydrogen peroxide; NAS, *N*-acetylserotonin; 2-HM, 2-hydroxymelatonin; AFMK, *N*^1^-acetyl-*N*^2^-formyl-5-methoxyknuramine; * *p* < 0.05; ** *p* < 0.01; ns, not significant by Duncan’s test.

**Table 2 ijms-21-00782-t002:** Output of two-way ANOVA for the interactive effects of drought stress and nano-ZnO on the gene relative expression in maize.

	Nano-ZnO	Water Regime	Nano-ZnO × Water Regime
*Cu/Zn SOD*	**	**	**
*Fe/Mn SOD*	ns	**	**
*CAT*	**	ns	**
*APX*	**	**	ns
*TDC*	*	**	**
*ASMT*	ns	ns	**
*COMT*	*	**	**
*SNAT*	**	*	ns

* *p* < 0.05; ** *p* < 0.01; ns, not significant by Duncan’s test.

**Table 3 ijms-21-00782-t003:** Pearson’s correlation coefficients between the tested parameters in maize.

	Soluble Sugar	Proline	Soluble Protein	MDA	APX	SOD	CAT	H_2_O_2_	Try	Tryptamine	Serotonin	NAS	MT
Soluble sugar	1	0.22	0.472	0.297	0.476	0.667*	0.54	−0.199	0.57	−0.075	0.671 *	0.071	0.409
Proline		1	−0.443	0.371	0.973**	0.707*	0.441	0.560	0.458	−0.831 **	−0.094	0.734 *	0.037
Soluble protein			1	−0.544	−0.299	0.172	0.44	−0.917 **	0.469	0.752 **	0.600 *	−0.239	0.519
MDA				1	0.476	0.315	−0.093	0.700 *	−0.077	−0.739 **	−0.276	0.052	−0.284
APX					1	0.828 **	0.426	0.543	0.348	−0.787 **	0.245	0.399	0.255
SOD						1	0.673*	0.156	0.539	−0.422	0.501	0.218	0.492
CAT							1	−0.240	0.659*	−0.025	0.339	0.431	0.290
H_2_O_2_								1	−0.311	−0.857**	−0.403	0.260	−0.451
Try									1	−0.06	0.43	0.204	0.55
Tryptamine										1	0.28	−0.407	0.137
Serotonin											1	−0.262	0.627 *
NAS												1	−0.328
MT													1

MDA, malondialdehyde; SOD, total superoxide dismutase; POD, peroxidase; CAT, catalase; H_2_O_2_, hydrogen peroxide; Try, tryptophan; NAS, *N*-acetylserotonin; MT, melatonin; * *p* < 0.05; ** *p* < 0.01.

**Table 4 ijms-21-00782-t004:** Primers used in this study.

Primers	Sequences
*ZmActin*-F	5′-GCCGATCGTATGAGCAAGGA-3′
*ZmActin*-R	5′-CCACCGATCCAGACACTGTA-3′
*Cu/Zn SOD*-F	5′-GTGCATATCGACAGGACCAC-3′
*Cu/Zn SOD*-R	5′-CAATGGTTGCCTCGGCTATG-3′
*Fe/Mn SOD*-F	5′-GGCGGTCATGTGAACCATTCAATC-3′
*Fe/Mn SOD*-R	5′-CGCCTTCTGCATTCATCCTC-3′
*APX*-F	5′-GAGCGATCAGGACATTGTTG-3′
*APX*-R	5′-CTTTGTCACTTGGGAGCTGAAG-3′
*CAT*-F	5′-CAGATTGCTTTCTGCCCAGC-3′
*CAT*-R	5′-GCATCAGATAGTTTGGACCAAGGC-3′
*TDC*-F	5′-CATGCTCGTCACGCTTGTCTC-3′
*TDC*-R	5′-GCCTTGAAGAAGGTGGAGTG-3′
*ASMT*-F	5′-CTGGACCTCTCACACGTCGTC-3′
*ASMT*-R	5′-CCCAGTCATGCAGAATCCACTTG-3′
*COMT*-F	5′-GATCCACTTCATGAGGATGGC-3′
*COMT*-R	5′-CCCAGTCATGCAGAATCCACTTG-3′
*SNAT*-F	5′-CATCGTGCTCTACGCCGAGAC-3′
*SNAT*-R	5′-GGACTTGCGGTAGTATGCCATC-3′

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
