# Peer review of "Nano-ZnO-Induced Drought Tolerance Is Associated with Melatonin Synthesis and Metabolism in Maize"

_ijms, 2020, doi:10.3390/ijms21030782_

Round 1

Reviewer 1 Report

The quality of the revised version is significantly improved and it is acceptable.

Author Response

Response to Reviewer 1 Comments

Point 1: English language and style are fine/minor spell check required. The quality of the revised version is significantly improved and it is acceptable.

Response 1: Thanks.

Reviewer 2 Report

Dear Authors,

Presented resubmitted manuscript is improved but there is a small problem with figures legend. Pleas include the abbreviation development in these legends where it's needed. You should include the same abbreviation development as in legend for figure 2. I mean ZnO, N, W and D in others legends it greatly improve the reading of the figures and data. After this correction I accept the paper for publication.

Author Response

Response to Reviewer 2 Comments

Point 1: Presented resubmitted manuscript is improved but there is a small problem with figures legend. Please include the abbreviation development in these legends where it's needed. You should include the same abbreviation development as in legend for figure 2. I mean ZnO, N, W and D in others legends it greatly improve the reading of the figures and data. After this correction I accept the paper for publication.

Response 1: Thanks for the suggestions and comments. The explanations of abbreviation have been added in other figure legends.

Reviewer 3 Report

General comments: several important points need to be addressed, both in the methods section and in the discussion, that seem to be quite weak. Major revision in suggested.

Discussion:

The importance of chloroplast and mitochondrion that is quite elucidated in the introduction is not strong enough in the discussion. It seems that authors made are just commenting the results.

Literature is definitely rich in term of information related to potential mechanisms affecting the mitochondrial and chloroplast functionality. Not only considering ZnO but also other metal oxides with relatively similar characteristic.

Methods:

Paragraphs 4.1, 4.2. Did authors consider 1000 mg/L a realistic scenario of exposure? Are there any measurable particle dissolution, agglomeration, or precipitation at this concentration?

Why the Zeta potential and DLS have been measured in 100 mg/L suspension instead of the same concentration utilized for the study?

Paragraph 4.10. Please transform in a table the paragraph which contains the primers couples used for the study. Furthermore, PCR cycle and primers dissociation are missing, considering that SYBR green has been used for the study.

Figures

Figure 5 caption. Line 210, please correct the sentence “Data are reported as mean…”

Author Response

Response to Reviewer 3 Comments

Point 1: Moderate English changes required. Several important points need to be addressed, both in the methods section and in the discussion, that seem to be quite weak. Major revision in suggested.

Response1: Thanks for the suggestions and comments. The Methods and Discussion sections have been improved.

Point 2: The importance of chloroplast and mitochondrion that is quite elucidated in the introduction is not strong enough in the discussion. It seems that authors made are just commenting the results.

Response 2: Thanks for the comments. The discussion on the effects of nano-ZnO on mitochondria and chloroplast has been promoted.

Point 3: Literature is definitely rich in term of information related to potential mechanisms affecting the mitochondrial and chloroplast functionality. Not only considering ZnO but also other metal oxides with relatively similar characteristic.

Response 3: The mechanisms affecting the mitochondrial and chloroplast functionality by other metal oxide nanoparticles, such as nano-TiO2 have been added.

Point 4: Paragraphs 4.1, 4.2. Did authors consider 1000 mg/L a realistic scenario of exposure? Are there any measurable particle dissolution, agglomeration, or precipitation at this concentration?

Response 4: The concentration of nano-ZnO that we used was 100 mg L-1. For the application of nano-ZnO suspension to the pots, a total amount of 1000 mL nano-ZnO (100 mg L-1) suspension was added in three times.

Point 5: Why the Zeta potential and DLS have been measured in 100 mg/L suspension instead of the same concentration utilized for the study?

Response 5: The concentration of nano-ZnO used and measured for Zeta potential and DLS was 100 mg L-1.

Point 6: Paragraph 4.10. Please transform in a table the paragraph which contains the primers couples used for the study. Furthermore, PCR cycle and primers dissociation are missing, considering that SYBR green has been used for the study.

Response 6: We have transformed the primers couples to Table 4. The missing PCR program has been addressed.

Point 7: Figure 5 caption. Line 210, please correct the sentence “Data are reported as mean…”

Response 7: It has been corrected accordingly.

Round 2

Reviewer 3 Report

All the comments have been addressed.

This manuscript is a resubmission of an earlier submission. The following is a list of the peer review reports and author responses from that submission.

Round 1

Reviewer 1 Report

Dear Authors

Presented for review manuscript entitled: "Nano-ZnO induced drought tolerance is associated with melatonin synthesis and metabolism in maize" showed nice results according to improve the drought tolerance of maize. Generally I suggest to improve and correct English, check literature. Moreover I have few question to authors:

Did authors measure the level of any Reactive oxygen species? It would be nice to show level of ROS in relation to presented results. In my opinion authors should show some parameters that clearly indicate that maize treated with ZnO is drought tolerant or has improved drought tolerance. I mean some physiological parameters for example Fresh weight, dry weight, photosynthetic parameters etc. In relation to proline level it would be nice to show expression level of two main genes encoding crucial enzymes for proline synthesis and degradation P5CS and PDH and of course the activity of these enzymes.

Based on my remarks I accept the manuscript after minor revision.

Reviewer 2 Report

In the current study, the authors have reported that the Nano-ZnO induced drought tolerance is associated with melatonin synthesis and metabolism in maize. This is a well-designed and fully executed study. Much work has been done and the results are promising. It provides novel information on the melatonin research in plants, especially, as my knowledge, melatonin metabolite, AFMK is first reported to associated with abiotic stress in the plants. I only have some minor concerns on the current manuscript.

For each figure, the sample sizes (n = ?) should be identified. This information is also absent in the text. Without this information the readers cannot objectively evaluate the statistical analysis. In figure 9, the melatonin synthetic pathway, the arrows in the last step is in the wrong direction, please correct it. The new development as melatonin synthesis in plants should be discussed and the related references should be updated to further improve the quality of the report. For example, in plants, melatonin is synthesized in the mitochondria (Tan, D.-X. and Reiter, R.J. 2019. Mitochondria: the birth place, battle ground and the site of melatonin metabolism in cells. Melatonin Research. 2, 1 , 44-66. doi.10.32794/mr11250011.) and chloroplasts (Zheng X, Tan DX, Allan AC, Zuo B, Zhao Y, Reiter RJ, Wang L, Wang Z, Guo Y, Zhou J, Shan D, Li Q, Han Z, Kong J. Chloroplastic biosynthesis of melatonin and its involvement in protection of plants from salt stress. Sci Rep. 2017 Feb 1;7:41236. doi: 10.1038/srep41236.).

Reviewer 3 Report

I think that the major problem of the presented paper is a lack of control and questionable statistics. The authors did not provide data for control materials (regular size material). In order to demonstrate that the nano-sized materials were responsible for the observed toxic effects, the authors have to repeat all experiments with correspondent control (ZnO) in the same concentrations that were used for the nano-sized materials.  It is not clear if the biological effects of the tested nanomaterials related to their uptake by plants or if they acted through the ionic state. Thus, experiments with control materials have to be performed. Additionally, the number of replicates (3 pots) is absolutely not sufficient for plant-related experiments. The authors should use at least 8-10 pots. With such limited number of replicates, it is hard to believe in results. The characterization of nanoparticles was performed (Figure 1). It is good. But additionally, authors should check purity of used nanomaterials and not rely on data from commercial sources. The absence of contamination with endotoxin should be proved as well.